# Diagnostic efficacy of hand-held digital refractometer for determining total serum protein in indigenous sheep of Pakistan

**Madiha Sharif[1], Mushtaq Hussain Lashari**[1]*, **Umer Farooq**[2], **Musadiq Idris**[2], **Muhammad Abrar Afzal**[2]

1 Department of Zoology, The Islamia University of Bahawalpur, Bahawalpur, Pakistan, 2 Department of Physiology, The Islamia University of Bahawalpur, Bahawalpur, Pakistan

* mushtaq.hussain@iub.edu.pk

## Abstract

The study was designed to ascertain the diagnostic efficacy of hand-held digital refractometer in determining total protein (TP). The Sipli sheep (n = 128) were grouped as per gender (females = 99, males = 29) and age (G1 = up till 1 year, n = 35; G2 = from 1 to 2 years, n = 63; G3 = above 2 years, n = 30). The results regarding the overall mean (±SE) values for the TPs attained through serum chemistry analyzer (TP1) and hand-held digital refractometer (TP2) were non-significantly (P≥0.05) different (59.2±1.6g/L and 59.8±0.5g/L, respectively). However, the reference intervals (RIs) were quite different for the two TPs being 45.1–95.7g/L and 57.0–67.0g/L for TP1 and TP2, respectively. Similar results were seen for gender-wise and group-wise results. On the contrary, the results regarding correlation coefficient and loglinear regression showed a negative correlation between the two TPs (r = -0.0244) with an adjusted r-square of 0.059 (5.9% probability). Furthermore, the three tests implied to assess the level of agreement between the two methods (Cronbach alpha, Intra-class correlation coefficient, and Bland & Altman test) revealed least agreement between the two methods. In a nutshell, the results of TP through digital refractometer were not in concordance with those attained through serum chemistry analyzer. However, it can cautiously be used if these results are compared with relevant corrected RIs.

## Introduction

The diagnostic/prognostic tests performed out of the laboratory and near/at the site of patient are termed as point-of-care-tests (POCTs) as per the definition provided by the International Standard ISO 22870 [1]. These tests have a rapid turnaround time, and assist in rapid decision-making and faster care of the patient. In human medical practice, they are dubbed as bed-side tests, near-patient testing, and patient-focused testing; whereas in veterinary medical diagnostics, they are termed as cow-side tests, on-farm tests, barn-side tests or flock-side tests [2, 3]. On human medical side, there are various POCTs presently in vogue for detecting glucose, hemoglobin, blood gases, electrolytes, cardiac enzymes and drug metabolites. In order to attain

**Data Availability Statement:** All relevant data are within the manuscript.

**Funding:** The author(s) received no specific funding for this work.

**Competing interests:** The authors have declared that no competing interests exist.

reliable results of surveillance, outbreak or analytic testing from the POCTs, it is vital that they are kept under strict regulatory oversight, quality control, and periodic comparison/validation through gold-standard techniques. Regarding veterinary medical diagnostics, there is still a paucity of devising and validating POCT devices for field use. It has been estimated that the market of veterinary POCTs was around 2.15 billion dollars with a prediction of 12.3% growth during 2021 to 2030 [4]. Evolving technologies and increased market needs are the key reasons behind this escalatory pattern of POCTs in order to efficiently respond the new and emerging diseases.

Since their introduction in 1960s, the hand-held digital refractometers are being used extensively as POCT devices for determining Brix%, salinity, specific gravity, and total proteins (TP) in various bodily fluids. These instruments measure the angle of refraction between air and an aqueous solution, providing rapid and inexpensive determination of solutes in the fluids. These devices have widely and successfully been used in veterinary medical diagnostics for assessing failure in passive transfer (FPT) through determining IgGs in sows [5], dogs [6] and cattle calves [7, 8]. Similarly, they have also been used for determining TP in various species with varying sensitivity and specificity as compared to other gold-standard techniques [9–11]. A study conducted with an aim of determining serum TP in sheep through biuret method and refractometrically has reported statistical accuracy of 93.05% [11]. However, to the best of knowledge, the diagnostic efficacy of hand-held digital refractometers in comparison to the serum chemistry analyzers, for determining TP in indigenous sheep of Pakistan, has not yet been unearthed.

There are about 17 indigenous sheep breeds being reared in Pakistan in different geo-ecological zones. It has been claimed that these breeds probably originated from urial (*Ovisvignei*), the wild sheep of Afghanistan, Baluchistan and Central Asia [12]. Sipli breed of sheep is a thin-tailed indigenous sheep breed of Pakistan with a relatively long tail. Small number of this breed (n = 260) is being maintained at two institutional farms in South Punjab, Pakistan. It is a medium-sized sheep breed with an average body weight of 32.8kg for males and 29.2kg for females, and has a daily milk yield of 0.2–0.4 L [12, 13]. It has white body coat with white or light brown head/ears. Its head is medium sized and has a flat nose with ears reaching about 15 cm long [14]. It is mostly reared for mutton and wool purposes by the nomadic herders of Bahawalpur, Bahawalnagar and Rahim-Yar-Khan- the three cities which lay in the middle of the Cholistan desert, (Southern Punjab) Pakistan. As it is reared within the domains of the desert, distances far flung from metropole laboratories, hence their diseases mostly go unchecked as the herders cannot bring their sick animals/samples to the laboratories in time. Considering this constraint, the present research work focuses on the use of on-field, hand-held, digital refractometers for determination of TP- an analyte which is highly indicative of sheep health. The research work is the first of its kind being reported for this sheep breed with an objective to ascertain the diagnostic efficacy of hand-held digital refractometer in determining TP as compared to that attained through serum chemistry analyzer. It is hypothesized that the digital refractometer will give satisfactory results at par with those of serum chemistry analyzer.

## Materials and methods

### Geo-location of study

The present study was carried out simultaneously at the Livestock Farm, Faculty of Veterinary and Animal Sciences (FV&AS), The Islamia University of Bahawalpur (IUB), and Post-graduate Lab of Physiology, IUB, Pakistan. Both of these study venues are located close to each other (at distance of 1.5 km) within the premises of the university. The climate of Cholistan desert is arid and semi-arid tropical; the average temperature of Cholistan desert is 28.33˚C, average rainfall of Cholistan desert is up to 180mm [15].

## Experimental animals

The Sipli breed of sheep (n = 128) being reared at the Livestock Farm of FV&AS, IUB, Pakistan under intensive farming system were incorporated in the present study. The animals under study were grouped as per gender (females = 99, males = 29) and age (G1 = up till 1 year, n = 35; G2 = from 1 to 2 years, n = 63; G3 = above 2 years, n = 30). The animals are sent for grazing early morning. In the evening the feeding of animals includes fresh-cut and chopped seasonal fodder along with concentrate ration containing about 15% crude protein. In addition, maize silage and wheat straw is offered depending on need as and when required. The fresh clean drinking water remains available all the time. The animals have been assigned tag numbers in order to collect data.

## Ethics statement

The research work was approved by the Departmental Research Ethics Committee, Department of Physiology, IUB, Pakistan vide Letter No PHYSIO-77/2023-104 dated 13-11-2023.

## Blood collection and analyses

Approximately 5mL blood sample were collected from each experimental animal. Bleeding was conducted once with a total of 128 blood samples. The blood was collected aseptically from the jugular vein using a 5mL disposable syringe in yellow-capped vacutainers containing silica and a polymer gel for serum separation. The vacutainers were centrifuged at 3000rpm for 15 minutes by centrifuge machine (Centrifuge 800, China) for serum extraction. Serum was extracted in Eppendorf tubes, appropriately labeled, and transported in ice-packs to the Post-graduate Lab Physiology, IUB (situated at 1.5km from the sample collection site) for further analyses. Samples were visually inspected and confirmed for the level of hemolysis.

The serum TP from attained serum samples was analyzed within 2–3 hrs by the same person, using following two methods:

a. **Through serum chemistry analyzer:** The TP was determined through commercially available TP kit (Bioactive Diagnostic Systems, JTC, Cat No. 5–172, Germany) using semi-automated serum chemistry analyzer (Rayto-9600, China). The analyzer works on two basic principles *i.e.* optical and electrochemical techniques. As per the instruction manual of the commercial kit, the analyzer was set at 25˚C. The commercial kit had a sensitivity/limit of quantification 0.17g/dL (7.3g/L) and Linearity of up to 15g/dL (150g/L), and implied the working principle of biuret reaction according to which the protein form colored complex with cupric ions in alkaline medium. The TP thus attained was termed as TP1 in this study.

b. **Through digital refractometer:** Determination of TP was carried out using a temperature-compensated hand-held digital refractometer (Serum Protein Tester, DR503, China) as per the instructions of the manufacturer. Briefly, after turning on the device, it was cleared with the distilled water and the sample plate was dried. Distil water was used to confirm the functioning of the refractometer before the measuring of TP for the samples. A 0.2–0.3 mL sample was dripped on the plate, and the cover/lid was closed. Pressing the 'read' button one time provided the TP in g/dL. The set of readings was attained at room temperature *viz.* 25-26˚C. All the samples were analyzed twice and mean was determined as a final measure. The device had a performance range of 2.0–14.0g/dL, accuracy of ±0.2, and a resolution of 0.1. The TP thus attained was termed as TP2 in this study.

## Statistical analyses

Statistical Package for Social Sciences (Windows Version 12, SPSS Inc, Chicago, IL, USA) was used for data analyses. Normality of the data and homogeneity of variance were tested through Shapiro-wilk test and Levene's test, respectively. Mean (±Standard Error, SE) values for serum TP attained through serum chemistry analyzer (TP1) and through digital hand-held refractometer (TP2) were analyzed for difference through independent t-test. Pearson's correlation coefficient and linear regression were implied to assess the level of relation, and to deduce the regression prediction equation between the two values. Three tests were implied to check the level of agreement between the two types of tests *viz*. Bland & Altman, Cronbach Alpha and Intraclass Correlation Coefficient (ICC). The mean (±SE), median, range and reference intervals (RIs) ($25^{th}$ to $95^{th}$ percentile) were deduced for the data (n = 128) keeping in view the guidelines provided by the American Society for Veterinary Clinical Pathology [16] using the Reference Value Advisor (freeware v.2.1: http://www.biostat.envt.fr/reference-value-advisor).

## Results

The results regarding overall as well as group-wise data for RIs of the TP1 (attained through serum chemistry analyzer) and TP2 (attained through hand-held digital refractometer) are given in Table 1. Both the mean (±SE) values (for TP1 and TP2) were non-significantly (P≥0.05) different (as ascertained through independent t-test) with mean values of 59.2±1.6g/L and 59.8±0.5g/L, respectively. However, the RIs were quite different between the two TPs being 45.1–95.7g/L and 57.0–67.0g/L for TP1 and TP2, respectively. Similar results were seen for gender-wise and group-wise results being non-significantly (P≥0.05) different between all study groups, but with wide range of RIs.

The results regarding correlation coefficient and logilinear regression (Fig 1) between TP1 and TP2 showed a negative correlation between the two attributes (r = -0.0244) and an adjusted r-square of 0.059 (5.9% probability) with following regression prediction equation:

$$y = -0.073x + 64.17.$$

**Table 1. Reference intervals for serum total protein (g/dL) attained through serum chemistry analyzer (TP1) and through digital refractometer (TP2) as affected by sex and age in Sipli sheep (n = 128).**

| Groups | Mean±SE | Median (IQR) | Range (Min-Max) | RI ($25^{th}$ to $95^{th}$) | 95% CI |
|---|---|---|---|---|---|
| **TP1** | | | | | |
| Females (n = 99) | 59.3±1.8 | 58.2(28.2) | 78.3(21.2–99.5) | 44.7–95.8 | 55.6–63.1 |
| Males (n = 29) | 58.7±3.2 | 58.9(21.7) | 70.6(25.0–95.7) | 45.7–93.1 | 52.1–65.4 |
| G1 (n = 35) | 58.1±3.1 | 59.1(29.2) | 76.5(21.2–97.7) | 44.6–96.5 | 51.6–64.6 |
| G2 (n = 63) | 58.9±2.4 | 57.8(26.6) | 77.2(22.3–99.5) | 43.1–95.7 | 54.0–63.8 |
| G3 (n = 30) | 61.0±2.9 | 58.5(28.8) | 61.3(34.4–95.8) | 45.8–91.7 | 54.9–67.1 |
| **Overall (n = 128)** | **59.2±1.6** | **58.5(26.3)** | **78.3(21.2–99.5)** | **45.1–95.7** | **56.0–62.5** |
| **TP2** | | | | | |
| Females (n = 99) | 59.4±0.6 | 60.0(7.0) | 47.0(34.0–81.0) | 56.0–67.0 | 58.2–60.6 |
| Males (n = 29) | 60.9±0.7 | 61.0(6.0) | 17.0(51.0–68.0) | 58.5–67.5 | 59.4–62.4 |
| G1 (n = 35) | 59.8±1.3 | 61.0(8.0) | 47.0(34.0–81.0) | 57.0–72.2 | 57.1–62.5 |
| G2 (n = 63) | 59.3±0.6 | 60.0(8.0) | 22.0(47.0–69.0) | 56.0–66.8 | 58.1–60.5 |
| G3 (n = 30) | 60.6±0.6 | 60.0(4.0) | 17.0(51.0–68.0) | 58.9–67.4 | 59.2–62.0 |
| **Overall (n = 128)** | **59.8±0.5** | **60.0(6.7)** | **47.0(34.0–81.0)** | **57.0–67.0** | **58.8–60.7** |

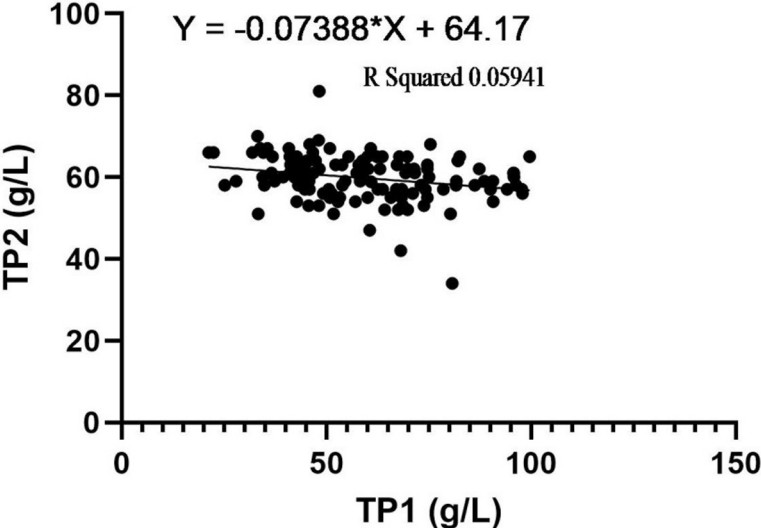

**Fig 1. Scatterplot for logilinear regression between TP1 (attained through serum chemistry analyzer) and TP2 (attained through hand-held digital refractometer).**

Results for Cronbach alpha and ICC between TP1 and TP2 are given in Table 2. Both the values for single measure and average values were lower between TP1 and TP2 being -0.135 and -0.313, respectively.

Similarly, Bland and Altman chart between TP1 and TP2 (Fig 2) showed a weak level of agreement. A proportional bias on the distribution of data around the mean difference line was noticed between TP1 and TP2 (Mean = 0.5; 95% CI = 39.8 to -40.9) with an S.D. of biasness being 20.58.

## Discussion

The present work is the first of its kind being reported for indigenous Sipli breed of sheep from Pakistan which was conducted with an aim to assess diagnostic efficacy of hand-held digital refractometer for determining serum TP, as compared to the serum chemistry analyzer which is considered as the gold-standard technique. The digital hand-held refractometer is globally acclaimed and in vogue, and its efficacy for determining TP has been elucidated. However, it failed to reveal satisfactory results for determining serum TP of the sheep in the present study as compared to those attained through serum chemistry analyzer which is obvious through the three tests of agreement implied in this study (Bland & Altman, Cronbach Alpha and ICC).

In the present study, the overall mean values of 59.2±1.6g/L and 59.8±0.5g/L attained through serum chemistry analyzer using commercial kit, and through hand-held digital refractometer, respectively are in line with those reported for three indigenous sheep of Iraq being

**Table 2. Cronbach alpha and intraclass correlation between TP1 (attained through serum chemistry analyzer) and TP2 (attained through hand-held digital refractometer).**

| TP1 vs TP2 | | | |
|---|---|---|---|
| **Intraclass Correlation** | | **95% CI** | **Cronbach Alpha** |
| Single Measure | -0.135 | -0.301–0.038 | -0.313 |
| Average Measures | -0.313 | -0.862–0.074 | |

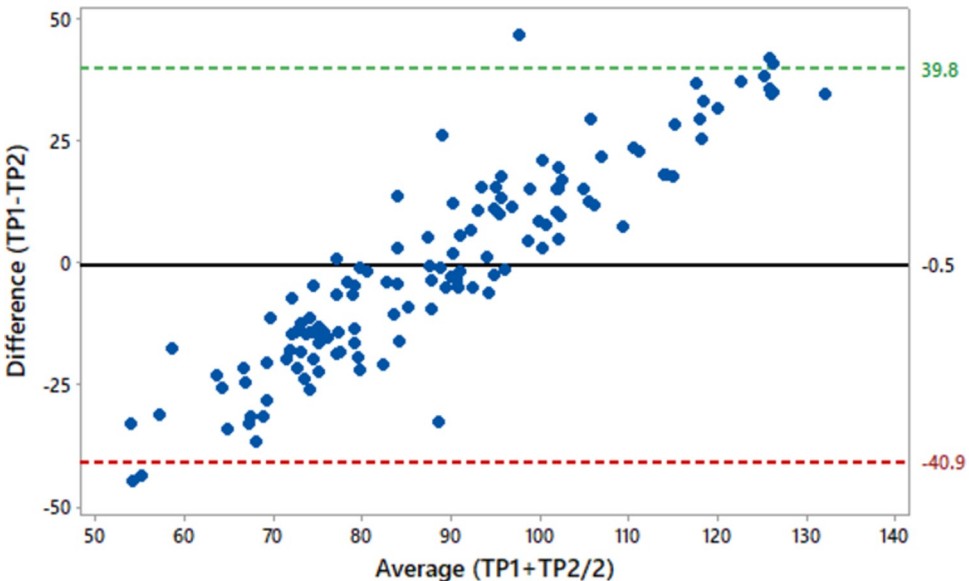

**Fig 2. Scatterplot of Bland and Altman test between difference of total protein determined through serum chemistry analyzer and through hand-held digital refractometer (TP1-TP2) and average of both TPs (TP1+TP2/ 2) in Sipli sheep (n = 128).** Black line indicates mean difference (-0.5) whereas the upper and lower dotted lines indicate upper (39.8) and lower (-40.9) values for 95% CI, respectively (SD of Bias 20.58).

62.0g/L [17]. Similar values of 58.2g/L have been reported for White Dorper and Suffolk sheep using Biuret method of TP determination [18]. Another study has also reported similar value of 60.0g/L using Biuret method for Dorper sheep of Brazil [19]. Higher values of 72.0g/L have been reported for free-ranging desert big-horn sheep [20] and Merino sheep [21] using automated serum chemistry analyzer. Similarly, a study from Greece has reported higher values of 83.3g/L and 79.1g/L for serum TP in sheep as determined through biuret method and through refractometry, respectively. Variation in breed, methods of sample handling and methods of TP determination may be attributed to these higher values.

Regarding the RIs (25th and 95th percentile) for the TP of sheep serum in the present study attained through two methods *i.e.* serum chemistry analyzer and hand-held digital refractometer, it was noticed that the RIs were quite different. They were 45.1–95.7g/L and 57.0–67.0g/L for TP1 and TP2, respectively. Comparing these RIs with those of prior studies, it was revealed that the RIs of the present study were though within the range provided in earlier studies [11, 21–23], yet different. As the RIs are different for two methods in our study, it seems inevitable that relevant RIs for each method may be considered while determining TP in sheep serum. The different RIs could be due to the different devices used in our study, their efficacy, their internal validation, and variability in temperature under which the samples were assayed.

Literature is rife with studies in which the digital refractometers (especially Brix refractometers) have successfully been used for determining serum immunoglobulins (IGs) in various species such as canines [6], bovines [7, 24, 25], porcine [26], ovine [27], and equine [28]. All these prior studies have validated hand-held digital refractometers for determining IGs in serum and have reported satisfactory sensitivity, specificity, negative predictive value and positive predictive values for this device dubbin it a reliable on-farm, cow-side POCT in contrast to our results. However, it is to be noted that the analyte(s) in these studies have been different (IGs) from that reported in present study (TP). Hence, the difference in used devices/

equipment, methodology, method sampling, processing protocols and species under consideration could be possible factors for this difference.

In the present study, though the independent t-test failed to reveal statistical difference between the two methods of TP determination, however other statistical tests of association/ agreement implied in this study presented a rather different picture. The correlation coefficient and regression showed a weak association between the two methods of TP determination (r = -0.0244, adjusted r-square = 0.059/5.9% probability) in the present study. This is in contrast to a previous study which has shown strong linear relation for both methods as ascertained through concordance correlation coefficient being 68.7% for sheep [11]. On similar lines, using Pearson's correlation coefficient, higher correlation has been reported for both methods in dogs (r = 0.632) and cats (r = 0.826) while determining TP [29].

The present study utilized three additional tests (Bland & Altman, Cronbach Alpha and ICC) for ascertaining level of agreement between the two methods using for determining TP i. e. through serum chemistry analyzer and hand-held digital refractometer. All these tests also indicated a poor level of agreement between the two studied tests. The Bland & Altman test is considered as a gold-standard test for checking level of agreement between two different techniques measuring one similar variable [30–32]. For the present study, this test revealed a proportional bias on the distribution of data around the mean difference line between TP1 and TP2 (Mean = 0.5; 95% CI = 39.8 to -40.9) with an S.D. of biasness being 20.58. These results are also not in line with those reported for sheep [11], dogs and cats [29]. Similarly, the ICC revealed negative values (-0.135 and -0.313 for single and average measures, respectively) for both tests in the present study. The difference in level of association between two methods of determining TP in our study could plausibly be attributed to the difference in temperature at which the samples were assayed, sensitivity and specificity of refractometers, and validation of analyzer.

Considering the results of the present study, it seems inevitable to conclude that the hand-held digital refractometer does not have substantial concordance to the results of a serum chemistry analyzer, and they cannot be used interchangeably. However, being an inexpensive and quick POCT device, its use cannot be overruled. Hence, it is recommended that while using a digital hand-held refractometer, comparison of results should be made with relevant corrected RIs. Further studies may be carried out with stricter sample handling and analyses protocols such as fixed temperature for both devices. Instead of visual assessment of hemolysis, the samples maybe assessed for hemolysis through appropriate chemistry analyzers in future.

## Author Contributions

**Conceptualization:** Mushtaq Hussain Lashari, Umer Farooq.

**Data curation:** Madiha Sharif.

**Formal analysis:** Musadiq Idris.

**Methodology:** Muhammad Abrar Afzal.

**Software:** Muhammad Abrar Afzal.

**Writing – original draft:** Umer Farooq.

**Writing – review & editing:** Madiha Sharif, Umer Farooq.

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
