## [Decision Letter · Decision Letter 0]

15 Feb 2024

PONE-D-23-37822Diagnostic efficacy of hand-held digital refractometer for determining total serum protein in indigenous sheep of PakistanPLOS ONE

Dear Dr. Lashari,

Thank you for submitting your manuscript to PLOS ONE. After careful consideration, we feel that it has merit but does not fully meet PLOS ONE’s publication criteria as it currently stands. Therefore, we invite you to submit a revised version of the manuscript that addresses the points raised during the review process.

We look forward to receiving your revised manuscript.

Kind regards,

Mahmud Iwan Solihin

Academic Editor

PLOS ONE

A clean copy of the edited manuscript (uploaded as the new *manuscript* file)”.

4. We note that your Data Availability Statement is currently as follows: [All relevant data are within the manuscript.]

Additional Editor Comments:

Please check the reviewer comments and adhere the suggestions.

Reviewers' comments:

Reviewer's Responses to Questions

**Comments to the Author**

1. Is the manuscript technically sound, and do the data support the conclusions?

Reviewer #1: Partly

Reviewer #2: Partly

2. Has the statistical analysis been performed appropriately and rigorously? 

Reviewer #1: I Don't Know

Reviewer #2: Yes

3. Have the authors made all data underlying the findings in their manuscript fully available?

Reviewer #1: Yes

Reviewer #2: Yes

4. Is the manuscript presented in an intelligible fashion and written in standard English?

Reviewer #1: No

Reviewer #2: Yes

5. Review Comments to the Author

Reviewer #1: Overview

This paper has the basis of a good publication, but needs significant adjustment to be publishable. More detail is needed about the materials and methods, including the sample handling, analysis implementation and statistical methods used, see Katsoulos et al (2017) and the references they cite for a demonstration of what is required.

The results differ significantly from those published by other authors, given that refractometer results appear to decrease with increasing serum chemistry TP results. While these results could be correct for this particular dataset, where such major deviation to expected outcomes occur, rigorous checking of the data and it’s construction is needed by the authors. This needs to include a review of the sample handling processes - timings and temperatures, and data handling. It would be recommended that the authors return to the raw data records (paper or whichever format they were first recorded in) and to check them against the final dataset and ensure that there hasn’t been a mistake in the sorting of data which has resulted in a loss of connection between the TP1 results and their relevant TP2 results.

As TP1 and TP2 are acquired by different methods, I would not expect them to give the same numerical result, but would expect them to be correlated, so that in clinical practice different reference ranges would be needed depending on which methodology was being used, and an theoretical equation can be produced to simulate the relationship between the outputs of these 2 methods (Katsoulos et al., 2017). Also, the main comparison of interest is how well correlated each pair of results are for each individual sample. Therefore I think that the data in table 1 and statistical comparison undertaken between the summary statistics for the whole group of results should moved to the end of the results section and given less attention in the results and a abstract, or removed altogether.

Also, the English language used needs to be reviewed before resubmission.

All abbreviations need to be given in full and then the abbreviation in parentheses the first time they are used in the abstract and in the full text (TP, RI, POCT, etc …). For example, ‘total protein (TP)…’

Abstract

The abstract is difficult to follow, it needs simplifying, giving the interpretation of the results and then results that support these interpretations in brackets.

M&M

More detail is needed about the sample handling (including timings and temperatures etc) and assessment of test precision, see for Katsoulos et al (2017) for examples.

Also, the method used by the serum chemistry analyser to test TP is not given. The authors need to find out from the manufacturers which method is used (biuret, refractometry, other) and state this.

How many times was each test run on each sample? Duplicate testing is advised, triplicate is better, but not necessary.

Were all samples checked for haemolysis etc.? Were any excluded for this?

Statistical analysis

Outliers - what was considered to be an outlier? Why were these removed? Any samples with TP levels that are physiologically viable should be included, because it is often the animals outside of the reference ranges that we are interested in.

More detail is needed in the statistical methods, to give more detail about how each of the tests were implemented

Results

Tables 2 and 3 should be combined into one table.

Discussion

Lines 180 and 181 - check the units for the TP values given here, the magnitude is x10 different to the other results given but the units (g/L) are the same.

Lines 188-190 - I do not understand what the authors are trying to say in this sentence. The values of TP1 and TP2 given are of very similar in magnitude and are also similar in magnitude to those reported from other studies, contrary to what this sentence appears to be saying.

Line 211 - these results cannot be generalised to all sheep, as this study only included 1 breed in specific circumstances and differs significantly from results from other studies, and these other studies need to be referenced.

Panagiotis D. Katsoulos, Labrini V. Athanasiou, Maria A. Karatzia, Nektarios Giadinis, Harilaos Karat-zias, Constantin Boscos, Zoe S. Polizopoulou. Comparison of biuret and refractometry methods for the serumtotal proteins measurement in ruminants. Vet Clin Pathol46/4 (2017) 620–624 DOI:10.1111/vcp.12532

Reviewer #2: The study presents a pertinent exploration of point-of-care testing (POCT) in veterinary medicine, particularly in remote or field settings, highlighting its significance. While commendable, the manuscript would benefit from a more robust interpretation of statistical analysis to strengthen its conclusions. Furthermore, expanding upon the interpretation of findings and discussing their clinical relevance would augment the study's impact on the veterinary medicine community. Addressing inherent limitations and suggesting avenues for future research are essential for a comprehensive understanding of the results. On top of that, enhancing the results and discussion section organization would augment readability and strengthen the scholarly integrity of the manuscript. With several adjustments (described below), the study holds promise for publication and warrants consideration.

Abstract:

The manuscript's abstract on the diagnostic efficacy of a hand-held digital refractometer for determining total serum protein in indigenous sheep of Pakistan presents the study's objectives, methods, and critical findings. However, it could benefit from a more straightforward presentation, interpretation of results, and simplification of the statistical analyses described. In addition, a more concise and focused presentation of the most significant results and a more precise interpretation of their implications for veterinary practice or research would enhance the abstract's readability and effectiveness.

Introduction:

Line 35–50: The introduction provides a general overview of POCTs in both human and veterinary medicine and briefly discusses hand-held digital refractometers for assessing various parameters in bodily fluids. However, it lacks a more thorough review of the existing literature on refractometers for determining TP in sheep. Incorporating relevant studies that have explored similar topics could help contextualize the current research within the broader scientific landscape and highlight any gaps or inconsistencies in the existing literature that the study aims to address.

Line 61–71: The introduction briefly mentions the indigenous Sipli breed of sheep in Pakistan and its significance for the local agricultural economy. However, providing a more detailed rationale for selecting this specific breed for the study would be beneficial. Why is it essential to assess TP levels in this particular breed, and what unique challenges or opportunities does studying indigenous sheep present compared to other breeds or species?

Line 71–73: A concise statement outlining the anticipated outcomes or hypotheses of the study would be beneficial to enhance the introduction. This would help orient readers and provide a roadmap for understanding the significance of the study findings.

Material and methods:

Line 112: The manuscript states that 13 outliers were removed from the data before analysis. However, the methodology for outlier removal is not clearly described. It is essential to provide details on how outliers were identified and justify their removal from the dataset. In addition, the impact of outlier removal on the results should be addressed to ensure transparency and reproducibility of the findings.

Line 113–117: Abbreviations like RI (Reference Interval) and SE (Standard Error) should be spelled out upon first use in the text. On line 113, it's mentioned: "Hence, RIs were deduced for remaining data (n=128) keeping in view the guidelines provided by the American Society for Veterinary Clinical Pathology". On line 117, it states: "Mean (±SE) values for serum TP attained through serum chemistry analyzer (TP1) and digital hand-held refractometer (TP2) were analyzed for difference through independent t-test". However, there's no initial explanation for RIs and SE.

Results:

Line 126–131: It states that the mean values of TP1 and TP2 were "non-significantly (P≤0.05) different." This statement is contradictory because it suggests that the difference between TP1 and TP2 is non-significant, but then it specifies a p-value threshold of 0.05, which is commonly used to determine significance. The P≤0.05 implies that the p-value is less than or equal to 0.05. In statistical interpretation, a p-value less than the significance level (0.05) indicates statistical significance, suggesting evidence to reject the null hypothesis (i.e., a significant difference between the groups being compared). Please explain regarding this condition.

Discussion:

Line 176: Despite finding non-significant differences in mean TP values between the two methods (TP1 from serum chemistry analyzer and TP2 from hand-held digital refractometer), it's essential to consider the clinical significance of these findings. While statistical significance is a crucial measure of whether differences observed in a study are likely due to chance, it doesn't necessarily reflect the practical importance or impact of those differences. Even minor differences in TP values can have significant clinical implications in veterinary diagnostics. Total protein levels are crucial indicators of an animal's health status, reflecting nutritional status, hydration status, and potential underlying medical conditions. Therefore, even if the differences in mean TP values between the two methods are statistically non-significant, they may still be clinically relevant if they fall outside an acceptable range or result in misclassification of animals' health status. Additionally, the discussion could explore potential factors contributing to the lack of significant differences in mean TP values between the two methods. This could include factors such as the sensitivity and specificity of the hand-held digital refractometer, variations in sample handling and processing, or inherent limitations of the analytical techniques employed.

Line 187: The wide discrepancies in reference intervals (RIs) between TP1 and TP2 raise significant concerns about the clinical utility and reliability of the hand-held digital refractometer for determining TP in sheep serum. Furthermore, the discussion should consider the limitations of the hand-held digital refractometer in accurately determining TP in sheep serum, particularly in comparison to the serum chemistry analyzer, considered the gold standard. Factors such as instrument calibration, sample volume requirements, and interference from other constituents in the serum could contribute to the discrepancies observed in reference intervals.

Line 194: It has been mentioned that digital refractometers (especially Brix refractometers) have successfully been used to determine serum immunoglobulins (IGs) in various species. Nevertheless, to provide a nuanced understanding of the diagnostic efficacy of hand-held digital refractometers, the discussion should explore potential factors contributing to variability in study findings across different studies. This could include differences in instrument calibration, sample handling, processing protocols, population demographics, and analytical methodologies. By examining these factors, the discussion can offer insights into the sources of variability in study outcomes and the broader implications for veterinary diagnostics.

Line 200–209: The discussion on agreement analysis, encompassing Bland & Altman, Cronbach Alpha, and Intraclass Coefficient, warrants a more thorough interpretation. Specifically, elucidating the implications of observed proportional bias in the Bland & Altman analysis and the significance of negative values in the Cronbach Alpha and Intraclass Correlation coefficients is crucial. While acknowledging potential biases in data collection, a deeper exploration of this issue and its potential impact on study findings is warranted. Addressing potential sources of bias or confounding factors that may have influenced results could offer valuable insights for interpreting findings comprehensively. Integrating these elements into the discussion enables a more robust understanding of the agreement analysis outcomes and their implications, enhancing the overall coherence and depth of the discussion section.

Line 210: The conclusion begins with a clear statement regarding the inadequacy of the hand-held digital refractometer for determining serum TP in sheep. However, the subsequent statement suggesting the potential use of other refractometer models with higher sensitivity and specificity is somewhat ambiguous. It's unclear whether the conclusion is definitive or speculative. To address this issue, the conclusion should clarify the rationale behind suggesting the exploration of other refractometer models. It should specify whether this recommendation is based on limitations identified in the current study, such as specific technical shortcomings of the hand-held digital refractometer used, or if it is merely speculative and based on the possibility that different models may yield different results. Providing additional context and justification for this recommendation would help readers better understand the implications of the study findings and the potential avenues for future research.

6. PLOS authors have the option to publish the peer review history of their article (what does this mean?). If published, this will include your full peer review and any attached files.

Reviewer #1: No

Reviewer #2: No

---

## [Author Response · Author response to Decision Letter 0]

17 Feb 2024

Dated: 17th of February, 2024

Dear Editor(s)

Kindly find below point-wise justification and actions taken on Manuscript # PONE-D-23-37822 as per the reviewer’s comments:

Reviewer’s Comments Action Taken/Justification

REVIEWER 1

More detail is needed about the materials and methods, including the sample handling, analysis implementation and statistical methods used, see Katsoulos et al (2017) and the references they cite for a demonstration of what is required. Keeping in view the reference provided by the esteemed reviewer (Katsoulos et al. 2017) and the instructions, details have been added in all relevant chapters of the amended draft, please. 

Where such major deviation to expected outcomes occur, rigorous checking of the data and its construction is needed by the authors. Done and reconfirmed, please. 

It would be recommended that the authors return to the raw data records (paper or whichever format they were first recorded in) and to check them against the final dataset and ensure that there has been no mistake in the sorting of data which has resulted in a loss of connection between the TP1 results and their relevant TP2 results. The data has been reconfirmed and it is reiterated that that the final data was soundly sorted and had no outliers or errors in it. Statistical analysis part of the amended draft elaborates the procedure, please. 

As TP1 and TP2 are acquired by different methods, I would not expect them to give the same numerical result, but would expect them to be correlated, so that in clinical practice different reference ranges would be needed depending on which methodology was being used, and an theoretical equation can be produced to simulate the relationship between the outputs of these 2 methods (Katsoulos et al., 2017). - The Pearsons correlation coefficient was implied which showed r-square of 0.059 (5.9% probability). This has been mentioned in the article, please. 

Furthermore, regression was implied and regression prediction equation was also deduced which has been incorporated in the amended draft, please. 

The English language used needs to be reviewed before resubmission. Revised accordingly, please. 

All abbreviations need to be given in full and then the abbreviation in parentheses the first time they are used in the abstract and in the full text Corrected at all relevant places in the amended draft, please. 

Abstract

The abstract is difficult to follow, it needs simplifying, giving the interpretation of the results and then results that support these interpretations in brackets. Abstract has been redone, please. 

M&M

- More detail is needed about the sample handling (including timings and temperatures etc) and assessment of test precision, see for Katsoulos et al (2017) for examples.

- Also, the method used by the serum chemistry analyzer to test TP is not given. The authors need to find out from the manufacturers which method is used (biuret, refractometry, other) and state this.

- How many times was each test run on each sample? Duplicate testing is advised, triplicate is better, but not necessary.

- Were all samples checked for hemolysis etc.? Were any excluded for this? 

- Appropriate details regarding the serum extraction, timing, and TP determination have been added, please.

- The optical and electrochemical techniques utilized by the analyzers has been mentioned in amended draft, please. 

- Yes, the samples were visually assessed for the level of hemolysis. None of the samples were hemolyzed enough to be discarded. Mentioned in M&M chapter accordingly, please. 

Statistical analysis

- Outliers - what was considered to be an outlier? Why were these removed? Any samples with TP levels that are physiologically viable should be included, because it is often the animals outside of the reference ranges that we are interested in.

- More detail is needed in the statistical methods, to give more detail about how each of the tests were implemented Initially, data of 141 sheep was attained and outliers were visually inspected as well as with Shapiro-wilk test. Outliers (values far beyond normal range) were deleted. 

However, in amended draft we have omitted the mention of outliers and the whole draft has been redevised with a data of 128 sheep/samples, please. Corrections have been made in appropriate sections. 

Statistical details have been made robust through appropriate addition in the amended draft, please. 

Results

Tables 2 and 3 should be combined into one table. - Combined as Table 1 in amended draft, please.

Discussion

- Lines 180 and 181 - check the units for the TP values given here, the magnitude is x10 different to the other results given but the units (g/L) are the same.

- Lines 188-190 - I do not understand what the authors are trying to say in this sentence. The values of TP1 and TP2 given are of very similar in magnitude and are also similar in magnitude to those reported from other studies, contrary to what this sentence appears to be saying.

- Line 211 - these results cannot be generalized to all sheep, as this study only included 1 breed in specific circumstances and differs significantly from results from other studies, and these other studies need to be referenced. 

- The units have accordingly been corrected. 

- The RIs in our study were different for both methods (TP1 and TP2). Though they were within the range reported in earlier studies. Corrections have accordingly been made in the amended draft, please. 

- Agreed that our conclusion was not definitive and clear. It has been rewritten with clarity in amended draft, please. 

REVIEWER 2

The manuscript would benefit from a more robust interpretation of statistical analysis to strengthen its conclusions. The discussion and conclusion have been redevised, please. 

Expanding upon the interpretation of findings and discussing their clinical relevance would augment the study impact on the veterinary medicine community. 

Addressing inherent limitations and suggesting avenues for future research are essential for a comprehensive understanding of the results. On top of that, enhancing the results and discussion section organization would augment readability and strengthen the scholarly integrity of the manuscript. 

Abstract

A more concise and focused presentation of the most significant results and a more precise interpretation of their implications for veterinary practice or research would enhance the abstracts readability and effectiveness. Abstract has been rewritten, please. 

Introduction

Line 35&50: The introduction provides a general overview of POCTs in both human and veterinary medicine and briefly discusses hand-held digital refractometers for assessing various parameters in bodily fluids. However, it lacks a more thorough review of the existing literature on refractometers for determining TP in sheep. Incorporating relevant studies that have explored similar topics could help contextualize the current research within the broader scientific landscape and highlight any gaps or inconsistencies in the existing literature that the study aims to address. 

Line# 58-60

Literature regarding TP determination in sheep serum has been added in Introduction, please. 

Line 61&71: The introduction briefly mentions the indigenous Sipli breed of sheep in Pakistan and its significance for the local agricultural economy. However, providing a more detailed rationale for selecting this specific breed for the study would be beneficial. Why is it essential to assess TP levels in this particular breed, and what unique challenges or opportunities does studying indigenous sheep present compared to other breeds or species? Line# 75-82

Te Sipli sheep breed is being reared under pastoral nomadism in the Cholistan desert, Pakistan. Its herds are mostly far away from the metropole laboratories and this constraint makes it inevitable that on-field POCT devices (such as hand-held refractometers) may be validated. The reason has clearly been justified in amended draft, please. 

Line 71&73: A concise statement outlining the anticipated outcomes or hypotheses of the study would be beneficial to enhance the introduction. This would help orient readers and provide a roadmap for understanding the significance of the study findings. The statement of hypothesis is added in amended draft, as directed, please. 

M & M

Line 112: The manuscript states that 13 outliers were removed from the data before analysis. However, the methodology for outlier removal is not clearly described. It is essential to provide details on how outliers were identified and justify their removal from the dataset. In addition, the impact of outlier removal on the results should be addressed to ensure transparency and reproducibility of the findings. 

- Initially, data of 141 sheep was attained and outliers were visually inspected as well as with Shapiro-wilk test. Outliers (values far beyond normal range) were deleted. 

However, in amended draft we have omitted the mention of outliers and the whole draft has been redevised with a data of 128 sheep/samples, please. Corrections have been made in appropriate sections. 

Statistical details have been made robust through appropriate addition in the amended draft, please. 

Line 113&117: Abbreviations like RI (Reference Interval) and SE (Standard Error) should be spelled out upon first use in the text. there's no initial explanation for RIs and SE. - Corrected and the the mean (±SE), median, range and reference intervals (RIs) (25th to 95th percentile) were deduced for the data (n=128) keeping in view the guidelines provided by the American Society for Veterinary Clinical Pathology using the Reference Value Advisor. Similar has been mentioned in amended draft, please.

Results

Line 126&131: It states that the mean values of TP1 and TP2 were non-significantly (P0.05) different. This statement is contradictory because it suggests that the difference between TP1 and TP2 is non-significant, but then it specifies a p-value threshold of 0.05, which is commonly used to determine significance. The P0.05 implies that the p-value is less than or equal to 0.05. In statistical interpretation, a p-value less than the significance level (0.05) indicates statistical significance, suggesting evidence to reject the null hypothesis (i.e., a significant difference between the groups being compared). Please explain regarding this condition. 

- We apologize that the symbol of ≤ was inserted instead of ≥. 

In fact, as per the independent t-test, the mean values of both TPs were non-significantly different (P≥0.05) indicating that both methods can be used interchangeably. 

However, interestingly when the tests of agreement were implied (Cronbach alpha, Intraclass coefficient and Bland-Altman), less agreement between two methods was revealed. 

Corrections have accordingly been made in amended draft, please. 

Discussion

The discussion could explore potential factors contributing to the lack of significant differences in mean TP values between the two methods. This could include factors such as the sensitivity and specificity of the hand-held digital refractometer, variations in sample handling and processing, or inherent limitations of the analytical techniques employed.

The discussion has been made stronger with appropriate justifications and additions of newer references, please. 

Line 187: The wide discrepancies in reference intervals (RIs) between TP1 and TP2 raise significant concerns about the clinical utility and reliability of the hand-held digital refractometer for determining TP in sheep serum. Furthermore, the discussion should consider the limitations of the hand-held digital refractometer in accurately determining TP in sheep serum, particularly in comparison to the serum chemistry analyzer, considered the gold standard. Factors such as instrument calibration, sample volume requirements, and interference from other constituents in the serum could contribute to the discrepancies observed in reference intervals. As the range of RIs was different for two different methods, in our study, hence we have recommended that relevant RIs may be considered while using relevant technique of TP determination. The corrections, as directed have accordingly been made in the amended draft, please.

Line 194: It has been mentioned that digital refractometers (especially Brix refractometers) have successfully been used to determine serum immunoglobulins (IGs) in various species. Nevertheless, to provide a nuanced understanding of the diagnostic efficacy of hand-held digital refractometers, the discussion should explore potential factors contributing to variability in study findings across different studies. This could include differences in instrument calibration, sample handling, processing protocols, population demographics, and analytical methodologies. The discussion has been rewritten keeping in view these factors mentioned by the esteemed reviewer, please.

Line 200&209: The discussion on agreement analysis, encompassing Bland & Altman, Cronbach Alpha, and Intraclass Coefficient, warrants a more thorough interpretation. Specifically, elucidating the implications of observed proportional bias in the Bland & Altman analysis and the significance of negative values in the Cronbach Alpha and Intraclass Correlation coefficients is crucial. While acknowledging potential biases in data collection, a deeper exploration of this issue and its potential impact on study findings is warranted. Addressing potential sources of bias or confounding factors that may have influenced results could offer valuable insights for interpreting findings comprehensively. Integrating these elements into the discussion enables a more robust understanding of the agreement analysis outcomes and their implications, enhancing the overall coherence and depth of the discussion section. 

Line 210: The conclusion begins with a clear statement regarding the inadequacy of the hand-held digital refractometer for determining serum TP in sheep. However, the subsequent statement suggesting the potential use of other refractometer models with higher sensitivity and specificity is somewhat ambiguous. It is unclear whether the conclusion is definitive or speculative. To address this issue, the conclusion should clarify the rationale behind suggesting the exploration of other refractometer models. It should specify whether this recommendation is based on limitations identified in the current study, such as specific technical shortcomings of the hand-held digital refractometer used, or if it is merely speculative and based on the possibility that different models may yield different results. Providing additional context and justification for this recommendation would help readers better understand the implications of the study findings and the potential avenues for future research. The conclusion has been rewritten keeping in view the directions, please. 

DR. MUSHTAQ HUSSAIN LASHARI

(Corresponding Author)

---

## [Editor Report · Decision Letter 1]

6 Mar 2024

PONE-D-23-37822R1Diagnostic efficacy of hand-held digital refractometer for determining total serum protein in indigenous sheep of PakistanPLOS ONE

Dear Dr. Lashari,

Thank you for submitting your manuscript to PLOS ONE. After careful consideration, we feel that it has merit but does not fully meet PLOS ONE’s publication criteria as it currently stands. Therefore, we invite you to submit a revised version of the manuscript that addresses the points raised during the review process.

Kindly provide the reviewers' comments and respective authors' responsein tabular form in such a way that the revisions were clear.==============================

We look forward to receiving your revised manuscript.

Kind regards,

Mahmud Iwan Solihin

Academic Editor

PLOS ONE

Journal Requirements:

**Additional Editor Comments:**

The authors have made quite significant response to the reviewers' comments. However, authors need to put in a neater way in such a

way that is easier to read and understand, e.g. in tabular format

-per comment of reviewer (left column) and respective response of the author (right column).

---

## [Author Response · Author response to Decision Letter 1]

7 Mar 2024

The document titled 'Response to Reviewers' contained tabulated form of comment-wise amendments made in the draft, please. And the said document has been uploaded separately please.

---

## [Editor Report · Decision Letter 2]

13 Mar 2024

Diagnostic efficacy of hand-held digital refractometer for determining total serum protein in indigenous sheep of Pakistan

PONE-D-23-37822R2

Dear Dr. Lashari,

We’re pleased to inform you that your manuscript has been judged scientifically suitable for publication and will be formally accepted for publication once it meets all outstanding technical requirements.

Kind regards,

Mahmud Iwan Solihin

Academic Editor

PLOS ONE

Additional Editor Comments (optional):

Reviewers' comments:

Authors have made revisions.

---

## [Editor Report · Acceptance letter]

18 Mar 2024

PONE-D-23-37822R2 

PLOS ONE

Dear Dr. Lashari, 

I'm pleased to inform you that your manuscript has been deemed suitable for publication in PLOS ONE. Congratulations! Your manuscript is now being handed over to our production team.

Kind regards, 

on behalf of

Dr. Mahmud Iwan Solihin 

Academic Editor

PLOS ONE